# SpGesture: Source-Free Domain-adaptive sEMG-based Gesture Recognition with Jaccard Attentive Spiking Neural Network

**Weiyu Guo**[1]    **Ying Sun**[1,*]    **Yijie Xu**[1]    **Ziyue Qiao**[2]    **Yongkui Yang**[3]    **Hui Xiong**[1,4,*]

[1]Thrust of Artificial Intelligence, HKUST (Guangzhou), China
[2]School of Computing and Information Technology, Great Bay University, China
[3]Shenzhen Institute of Advanced Technology, Chinese Academy of Sciences, China
[4]Department of Computer Science and Engineering, HKUST, Hong Kong SAR, China
`wguo395@connect.hkust-gz.edu.cn; yings@hkust-gz.edu.cn;`
`yxu409@connect.hkust-gz.edu.cn; zyqiao@gbu.edu.cn;`
`yk.yang@siat.ac.cn; xionghui@ust.hk`

## Abstract

Surface electromyography (sEMG) based gesture recognition offers a natural and intuitive interaction modality for wearable devices. Despite significant advancements in sEMG-based gesture recognition models, existing methods often suffer from high computational latency and increased energy consumption. Additionally, the inherent instability of sEMG signals, combined with their sensitivity to distribution shifts in real-world settings, compromises model robustness. To tackle these challenges, we propose a novel SpGesture framework based on Spiking Neural Networks, which possesses several unique merits compared with existing methods: (1) Robustness: By utilizing membrane potential as a memory list, we pioneer the introduction of Source-Free Domain Adaptation into SNN for the first time. This enables SpGesture to mitigate the accuracy degradation caused by distribution shifts. (2) High Accuracy: With a novel Spiking Jaccard Attention, SpGesture enhances the SNNs' ability to represent sEMG features, leading to a notable rise in system accuracy. To validate SpGesture's performance, we collected a new sEMG gesture dataset which has different forearm postures, where SpGesture achieved the highest accuracy among the baselines ($89.26\%$). Moreover, the actual deployment on the CPU demonstrated a latency below 100ms, well within real-time requirements. This impressive performance showcases SpGesture's potential to enhance the applicability of sEMG in real-world scenarios. The code is available at `https://github.com/guoweiyu/SpGesture/`.

## 1 Introduction

Surface electromyography (sEMG) is a sensing modality that decodes motor intentions from muscle electrical signals preceding movement to enable natural and intuitive interactions. It has distinct advantages in gesture recognition for real-time applications. Specifically, sEMG provides rich and comprehensive motion information, making it an excellent resource for accurate and efficient wearable gesture recognition [12]. Moreover, sEMG signals can emerge anywhere from 50 to 150 milliseconds prior to the actual motor activity, enabling the anticipation of movements.

In recent years, Spiking Neural Networks (SNNs) [41, 21, 20] provide an unparalleled chance for developing more practical and efficient sEMG-based gesture recognition systems. SNNs emulate

---

*Corresponding authors.

the spiking behavior of biological neurons with a unique binary information communication protocol [53]. This binary communication is particularly amenable to the architectural specifics of sparse neuromorphic hardware [50]. Besides, the primary computations in SNNs revolve around spike-based accumulate (AC) operations [13]. The event-driven nature of these networks [51, 23, 82] enables calculations to be made only when there is a change or 'event' in the input, thereby circumventing the need to process zero values. Therefore, compared to conventional Artificial Neural Networks (ANNs) that typically rely on energy-demanding multiply-and-accumulate (MAC) operations [64] and are normally deployed on high-computing-power hardware like GPUs, SNNs demonstrate substantially lower power consumption [42], positioning them as a promising candidate for developing energy-efficient gesture recognition systems [2, 71, 52].

Although SNNs are computationally efficient, they struggle to match the accuracy of ANN-based models [15]. In particular, the major problem is that the binary and sparse feature representations make it difficult to perform regular contiguous similarity computations. This limitation hinders expressive operations like attention mechanisms and advanced representation alignment algorithms, such as domain adaptation. For example, attention-based structures like the Transformer have demonstrated remarkable performance in Natural Language Processing [16, 40, 8, 69], Computer Vision [26, 30, 39, 70, 68], Time-Series Processing [76, 77, 75] and Decision-Making tasks[56, 57], leading to a wave of attention-centric architecture designs, underscoring the importance and versatility of attention mechanisms in deep learning. However, with the proportion of '1's typically less than $5\%$, the dot product in cosine similarity inherent to attention mechanisms tends to yield results close to zero [67]. Existing work often first converts spike signals into continuous values for similarity calculations, but this can increase the inference latency and energy consumption of SNNs. There is still a lack of methods for directly implementing advanced operations on spike features in SNN for realizing effective sEMG-based gesture recognition systems.

To tackle these challenges, we propose an SNN-based solution for a low-power yet accurate sEMG-based gesture recognition framework. Specifically, we first propose a novel Jaccard Attention Spiking Neural Network (JASNN) to enhance the representativeness of the network for sEMG features. In particular, different from existing studies that exploit attention to regulate membrane potentials and subsequently influence spiking activity [67], we propose a Spiking Jaccard Attention that calculates attention directly on spike sequences, which enables more straightforward computationally effective attention calculation under SNN schema. Indeed, such a computation process predominantly involves 'comparison' operations, aligning well with the design principles of neuromorphic chips and preserving the low-power properties of SNNs. Moreover, to address the distribution shift problem, we propose a novel Spiking Source-Free Domain Adaptation based on Membrane Potential Memory. Our method leverages the changing membrane potential curve as a memory list and uses it to generate pseudo-labels based on the $k$-nearest neighbors that are most similar to the current sample. In particular, we incorporate a random exploration mechanism to avoid overfitting during pseudo-label generation and bolster the model's generalizability. With our method, we achieve knowledge transfer without sharing the data, which enhances gesture recognition accuracy in an unlabeled environment under privacy reservation.

To better reflect real-world conditions, we collect a new sEMG-based gesture dataset that includes different forearm postures, acknowledging that variations in forearm posture can significantly influence the distribution of sEMG data. Our experimental results demonstrate that our algorithm not only significantly outperforms other SNN-based algorithms in gesture recognition accuracy but also matches the performance of state-of-the-art methods in the Deep Neural Networks (DNNs) category. Furthermore, the Spiking Jaccard attention method we proposed substantially enhances the accuracy of SNN algorithms. Regarding inference speed, Spiking Jaccard attention is **36.37x** faster on a CPU than traditional attention mechanisms. Our innovatively designed SSFDA method, which does not require source data or labels, improved the gesture recognition accuracy by **4.5%**. These results collectively underline the effectiveness and efficiency of our proposed approach in addressing the challenges in sEMG-based gesture recognition. Our contribution can be summarized as follows:

- We propose a Jaccard similarity-based attention mechanism specifically designed for SNNs. This innovative approach preserves the original computational characteristics of SNNs, boosts inference efficiency, and counteracts the accuracy degradation caused by sparse spiking sequences.
- To the best of our knowledge, we are among the first to propose an SNN-oriented SFDA algorithm. This enables users to capture gesture actions under one specific forearm posture and empowers the

model to unsupervised learning the features under other forearm postures, thereby bolstering its robustness during actual use.

- We collect a new sEMG-based gesture dataset that features a variety of forearm postures. This dataset can provide valuable resources for researchers aiming to develop robust gesture recognition algorithms for different forearm postures.

- The experimental results demonstrate performance improvements over state-of-the-art sEMG gesture recognition models, with particular benefits under varying forearm orientations. Our model also provides higher efficiency than existing attention schemes.

## 2 Related Works

**Spiking Neural Networks (SNNs)**, the third generation of neural networks, mimic biological neurons through binary spiking signals and handle temporal information effectively [19]. SNNs are energy-efficient, activating only a small portion of neurons during computation, unlike dense ANNs that rely on energy-intensive operations [2]. Neuromorphic chips like Tianjic [14], TrueNorth [1], and Loihi [13] exemplify this efficiency. Despite their energy advantages, SNNs have lower accuracy than DNNs due to sparse feature representation and simplistic structures. Attention mechanisms, widely used in DNNs [60], are under-explored in SNNs, posing challenges like spike degradation and gradient vanishing. Addressing these issues is crucial for improving SNN performance.

**Domain Adaptation (DA)** aims to leverage labeled source domain data to improve performance on unlabeled target domains, addressing domain shifts [62]. Traditional DA requires access to both source and target data, which is impractical in scenarios involving privacy or resource constraints [36]. **Source-Free Domain Adaptation (SFDA)** addresses this by adapting models without source data, crucial for privacy-sensitive applications like sEMG gesture recognition [22]. SFDA methods are categorized into data-centric and model-centric approaches [49]. Data-centric methods extend UDA techniques by reconstructing virtual domains or translating target data into source-style data [37]. Model-centric methods, like pseudo-labeling [35], entropy minimization [6], and contrastive learning [79], fine-tune models using target data. However, applying SFDA to SNNs is challenging due to their lower stability and sparse outputs.

## 3 Preliminaries

**Data Collection:** In human-computer interaction studies involving sEMG, diverse and representative data sets are crucial. Traditional research often collects sEMG data from a single forearm posture [4, 31], but variations in forearm posture significantly influence sEMG data distribution. Our methodology incorporates gestures performed in different forearm postures to better reflect real-world conditions. Participants were instructed to replicate gestures and forearm postures shown on a screen. Our dataset includes ten gestures across three forearm postures: P1 (forearm horizontal on a surface), P2 (forearm elevated diagonally with elbow anchored), and P3 (forearm horizontal). Each gesture was held for five seconds with a five-second relaxation period, repeated six times per posture. This approach aims to provide robust sEMG data reflecting practical variability. For further details on dataset collection, including information about the acquisition devices and specific measures taken, please refer to appendix A.1.

**Data Processing:** We used Root Mean Square (RMS) for initial feature extraction to enhance gesture recognition stability. RMS efficiently summarizes signal magnitude, indicating signal power. A 100ms time window with a 0.5ms step size captured transient sEMG characteristics, extracting features while maintaining high-resolution signal variations. RMS is further explained in appendix A.2.

## 4 Method

In the subsequent sections of this paper, we will present a sEMG-based gesture recognition solution with SNNs capable of handling distribution shifts. This solution can be divided into Jaccard Attention Spiking Neural Network (JASNN) and Spiking Source-Free Domain Adaptation (SSFDA). Firstly, we will introduce the unique SNN backbone JASNN deployed in our study. Following this, we will delve into our innovative design, the novel implementation of SSFDA within an SNNs framework – a first in the field. For more details about SNN, please refer to appendix A.3.

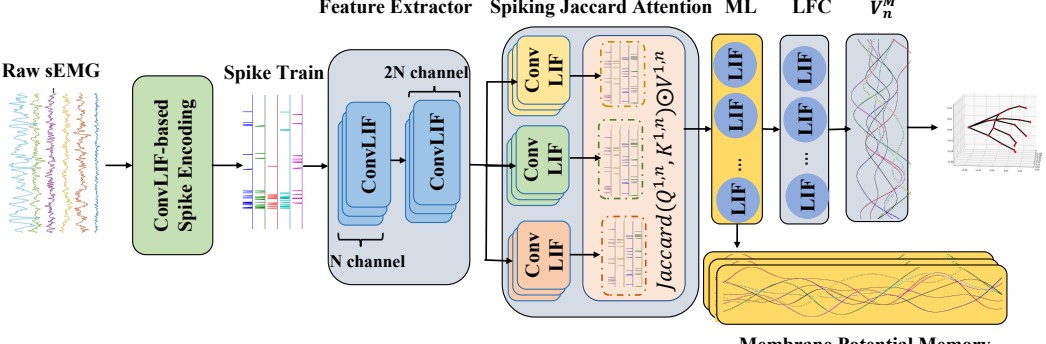

Figure 1: The pipeline of Jaccard Attention Spike Neural Network: Raw sEMG Data is first encoded into Spike Signals using ConvLIF. These signals pass through ConvLIF layers with $N$ and $2N$ channels. The processed data then goes through the Spiking Jaccard Attention mechanism.

## 4.1 Jaccard Attentive Spiking Neural Network

### 4.1.1 Network Overview

Our proposed Jaccard Attentive Spiking Neural Network (JASNN) comprises four primary components: a Convolutional Leaky Integrate-and-Fire (ConvLIF) based spike encoder and feature extractor, a Spiking Jaccard attention mechanism, LIF-based Classifier, and a membrane potential recording module. We detailed the first and third components in appendix A.4 A.6.

The ConvLIF-based spike encoding layer dynamically encodes sEMG signals into spike trains, capturing temporal dynamics effectively. The Multi-Channel ConvLIF extractor transforms these spikes into a higher-dimensional space for better feature representation. The Spiking Jaccard attention mechanism focuses on task-relevant features, enhancing meaningful information. The modified LIF layer translates spiking activity into classification results based on the highest membrane potential. Finally, the membrane potential recording module converts output spikes into membrane potentials for source-free domain adaptation.

### 4.1.2 Spiking Jaccard Attention

Attention mechanisms have enhanced DNNs in time-series analysis by focusing on important temporal aspects for better predictions. However, applying attention mechanisms to Spiking Neural Networks (SNNs) presents unique challenges. Firstly, SNNs' sparse neuron activation makes the dot product operation in attention mechanisms produce sparse spike trains, hindering learning due to reduced signal strength. Secondly, using the softmax function for attention scores increases computational complexity and energy consumption, which is unsuitable for SNNs' efficient processing requirements.

To address these concerns, we propose a novel Spiking Jaccard Attention (SJA) mechanism specifically designed for SNNs. As shown in Figure 2, unlike the method by Yao *et al.* [67], SJA can directly calculate the similarity on spike trains and retains the attention's query mechanism.

Given the binary nature of SNN layers outputs, the dot product approach in attention will make the feature too sparse. We introduce the SJA mechanism based on the Jaccard similarity, which is better suited for binary data. The Jaccard similarity between two sets A and B can be defined as:

$$\text{Jaccard}(A, B) = \frac{|A \cap B|}{|A \cup B|}. \tag{1}$$

Generally speaking, designing a spiking chip for SNNs mainly involves a large number of addition circuits and comparison circuits. Therefore, in the practical implementation of our proposed SJA, we retain the computational characteristics of the spiking chip to compute the Jaccard similarity more efficiently. This is achieved by calculating the intersection and union of the vectors using element-wise minimum and maximum operations, respectively. For two vectors $\mathbf{x}$ and $\mathbf{y}$, it can be described as:

$$\text{Jaccard}(\mathbf{x}, \mathbf{y}) = \frac{\sum_i \min(\mathbf{x}_i, \mathbf{y}_i)}{\sum_i \max(\mathbf{x}_i, \mathbf{y}_i) + \epsilon}. \tag{2}$$

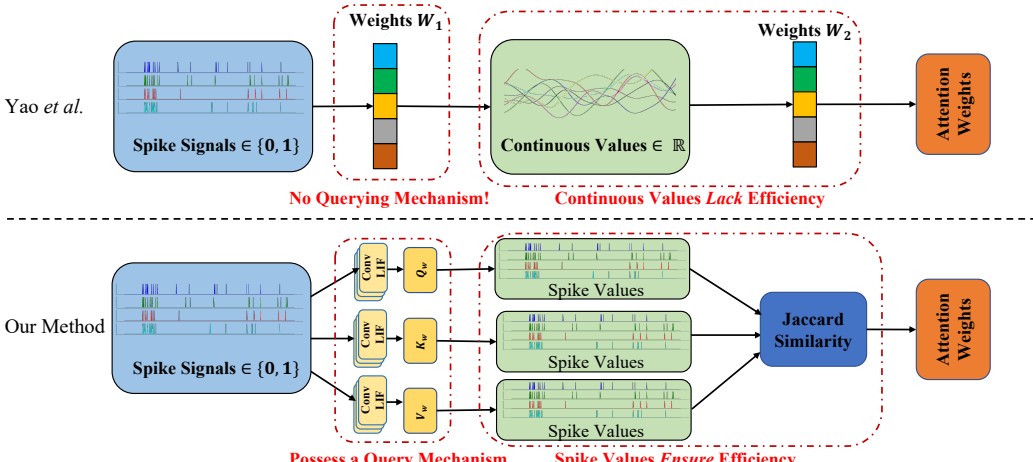

Figure 2: Comparison of MA-SNN and Spiking Jaccard Attention Modules. MA-SNN [67] uses fully connected layers with pooling but lacks a querying mechanism, leading to continuous intermediate values and lower efficiency. Our Spiking Jaccard Attention uses spike values for intermediate representations, enhancing efficiency and accuracy.

This approach enables efficient computation of the Jaccard similarity by taking advantage of the sparsity of the data in SNNs. By computing the sums over the element-wise minimum and maximum operations instead of using matrix dot multiplication operations, our algorithm becomes more easily deployable on Neuromorphic chips, thereby enhancing the computational efficiency of the attention mechanism within SNNs. We add a tiny constant to the denominator to avoid a division by zero when there are no spikes in the spike train.

So, we can modify the traditional attention formula by incorporating Jaccard similarity into the attention mechanism. The resulting SJA mechanism can be expressed as:

$$\text{SJA}(\mathbf{Q}, \mathbf{K}) = \frac{\sum_i \min(q_i, k_i)}{\sum_i \max(q_i, k_i) + \epsilon} \mathbf{V}, \tag{3}$$

where $\mathbf{Q}$, $\mathbf{K}$, and $\mathbf{V}$ represent the query, key, and value matrices, respectively, and $q_i$ and $k_i$ are the corresponding elements in the query and key matrices.

First, we consider the channel-wise uniform weighting method. This approach implies that the same weighting coefficient is applied to all elements along the channel dimension of $\mathbf{V}$. In this case, the attention weight is computed as a scalar, calculated by aggregating the elements within each channel: where $i$ is the index of the elements within the channel. The resulting scalar is then used as a weighting factor applied to each channel of $\mathbf{V}$:

$$\mathbf{V}_{\text{new}}[:, c, :] = \text{Jaccard}(\mathbf{Q}, \mathbf{K}) \cdot \mathbf{V}[:, c, :], \tag{4}$$

where $c$ represents the channel index. In this way, the values across all channels are scaled by the same weighting factor, thereby maintaining consistency across different channels.

Second, we consider the element-wise weighting method. In this case, the $\text{Jaccard}(\mathbf{Q}, \mathbf{K})$ result is computed independently for each element position $q, k$. This means that the attention weight for each element is obtained by calculating the value for the corresponding elements in $\mathbf{Q}$ and $\mathbf{K}$ at that position. These weights are then applied element-wise to $\mathbf{V}$:

$$\mathbf{V}_{\text{new}}[:, c, n] = \text{Jaccard}(\mathbf{Q}, \mathbf{K})[:, c, n] \cdot \mathbf{V}[:, c, n], \tag{5}$$

where $n$ represents the index along the sequence length. Under this element-wise weighting strategy, different positions within $\mathbf{V}$ are scaled independently based on their respective attention weights, which enables the model to capture finer-grained features.

The results presented in this paper are derived using the channel-wise weighting approach, as it is more suitable for the characteristics of sEMG data, and we did not validate the element-wise weighting approach due to these characteristics.

These two weighting strategies each have their respective applications: channel-wise uniform weighting is more appropriate for preserving feature consistency, while element-wise weighting is better suited for capturing localized differences. Depending on the computational complexity and the task requirements, an appropriate weighting strategy can be selected to achieve a balance between efficiency and performance.

The SJA mechanism leverages the sparsity of SNN outputs to significantly reduce computational complexity. Unlike traditional attention mechanisms with a complexity of $O(n^2 \cdot d)$, SJA focuses only on non-zero elements, resulting in a complexity of $O(b)$, where $b$ is the number of non-zero elements. This approach enhances computational efficiency and reduces energy consumption, as addition operations dominate SJA compared to the multiplication-heavy traditional attention, making SJA particularly advantageous for SNNs. Further complexity analysis can be found in appendix A.5.

## 4.2 Spiking Source-Free Domain Adaptation based on Membrane Potential Memory

The formal definition of the problem is as follows: given a labeled source domain $\mathcal{D}_s = \{(x_i^s, y_i^s)\}_{i=1}^{N_s}$, an unlabeled target domain $\mathcal{D}_t = \{x_j^t\}_{j=1}^{N_t}$ and a model $f_s$ trained on $\mathcal{D}_s$, the goal is to adapt or fine-tune the model $f_s$ such that its performance on the target domain $\mathcal{D}_t$ is optimized. The primary challenge stems from the different data distributions of the source and target domains, i.e., $P_s(x, y) \neq P_t(x, y)$, where $P_s$ and $P_t$ denote the data distributions of the source and target domains, respectively. In SFDA, the added complexity is that the source data $\mathcal{D}_s$ is not available when adapting or fine-tuning the model, while only having the source model $f_s$. Thus, the adaptation must rely on the properties and capabilities of the source model and unlabeled target data.

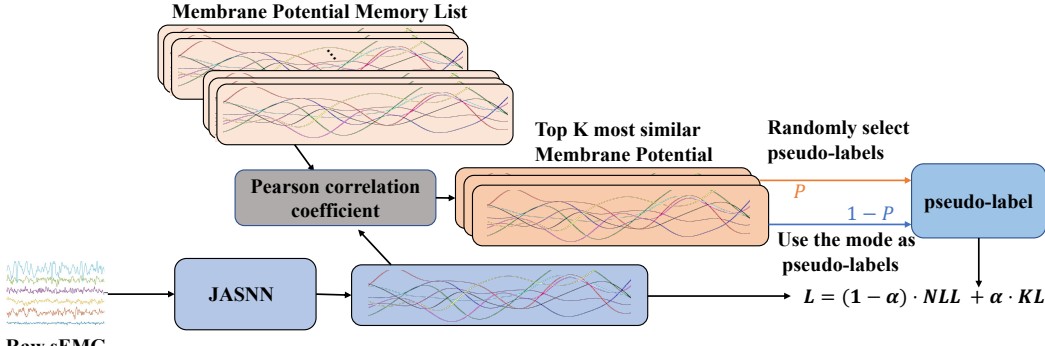

Figure 3: Computation flow of Spiking Source-Free Domain Adaptation. The process starts with selecting the $k$-nearest samples from the membrane potential memory using the Pearson correlation coefficient. Probabilistic Label Generation then produces pseudo-labels based on these $k$ samples. Gradients are computed with Smooth NLL and KL divergence loss. The membrane potential memory list is updated at each epoch's end.

Most previous methods consider similarity based on instance discrimination among all features in their loss functions, which can lead to high computational costs. This requirement can generate a significant computational overhead. In line with the approach taken by [65], we generate pseudo-labels using the $k$-most similar samples to the target sample with a consistency regularization. Furthermore, we introduce an exploration mechanism to mitigate overfitting. This strategy effectively maintains computational efficiency while enhancing the robustness and generalization of our SFDA approach.

Another challenge is that the intermediate layer features in SNNs are represented by Spike Trains, and existing methods for finding neighbors cannot directly compute them. To identify the semantically closest neighbors to a target domain sample, we utilize the membrane potential from the Memory Layer to construct a Membrane Potential Memory List. Note that we only use target source data to generate the Membrane Potential Memory List. The membrane potential encapsulates both spatial and temporal features, rendering it a more informative and efficient tool for our purpose. Membrane Potential Memory $M_n = \{V_{n,t}^m\}_{t=1}^T$ can be computed by:

$$V_{n,t}^m = S^t + \delta \cdot N(0, 1), \tag{6}$$

where $N(0, 1)$ represents Gaussian noise with mean 0 and standard deviation 1, and $\delta$ is a scaling factor. Integrating Gaussian noise with scaling offers two key benefits: regularization helps prevent overfitting, allowing the model to generalize better to unseen data, and noise introduction reduces the dominance of zeros in spike data, leading to a more balanced data representation. Figure 3 shows the SSFDA computation flow.

The core of the loss function is the alignment of predictions between the current target feature and its $k$-nearest neighbors in the Membrane Potential List, identified based on Pearson similarity. To achieve this, we introduce the following loss function Smooth Negative Log Likelihood (SNLL) Loss that combines two crucial components:

$$\mathcal{L} = -(1-\alpha)\frac{1}{n}\sum_{i=1}^{n}\sum_{k=1}^{K}\log\left(p(x_i)\cdot\text{argmax}\left(\mathcal{S}_k\right)\right) + \alpha\sum_{c=1}^{C}\text{KL}\left(\bar{p}_c\parallel q_c\right), \tag{7}$$

where

$$\mathcal{S}_k = \begin{cases} \text{mode}\left(\text{argmax}\left(\{\mathcal{M}\}_1^k\right)\right), & \text{with probability } (1-p), \\ \text{random}\left(\text{argmax}\left(\{\mathcal{M}\}_1^k\right)\right), & \text{with probability } p, \end{cases} \tag{8}$$

$$\{\mathcal{M}\}_1^k = \{\mathcal{F}_j \mid \text{top}K\left(\text{Pearson}\left(f(x_i), \mathcal{F}_j\right)\right), \mathcal{F}_j \in \mathcal{F}\}, \tag{9}$$

$$\bar{p} = \frac{1}{n}\sum_{i=1}^{n}p_c\left(x_i\right), q_c = \frac{1}{C}, \quad \text{for } c = 1, 2, \ldots, C. \tag{10}$$

The loss function, $\mathcal{L}$, is composed of two main terms: **Consistency Term:** The first component, $-(1-\alpha)\frac{1}{n}\sum_{i=1}^{n}\sum_{k=1}^{K}\log\left(p(x_i)\cdot\text{argmax}\left(\mathcal{S}_k\right)\right)$, is designed to advocate consistent predictions between a target feature and its $k$-nearest neighbors. It strives to minimize the negative logarithm of the inner product of the prediction score for the target sample, denoted by $p(x_i)$, and the aggregated prediction scores represented by $\text{argmax}\left(\mathcal{S}_k\right)$ of its $k$-nearest neighbors. $\mathcal{S}_k$ represents either the mode of the argmax values from the subset $\{\mathcal{M}\}_1^k$ with probability $p$, or a random selection from the same subset with probability $(1-p)$. By inducing similarity in predictions among closely related features, our model can discover latent structures and associations within the data; **Regularization Term:** The subsequent component, $\alpha\sum_{c=1}^{C}\text{KL}\left(\bar{p}_c\parallel q_c\right)$, uses the Kullback-Leibler divergence to measure the discrepancy between the model's average predicted class distribution, $\bar{p}_c$, and the ideal uniform distribution across classes, $q_c$. Specifically, $\bar{p}_c$ denotes the model's average prediction probability for class $c$ over all data samples. By comparing $\bar{p}_c$ with $q_c$, the divergence quantifies the deviation of the model's predictions from a perfectly balanced class distribution. The aim is to reduce the model's inclination to favor certain classes overly, ensuring a more balanced prediction landscape. In this configuration, The scalar $\alpha$ in the loss function acts as a balancing factor between predictive consistency and regularization.

### 4.3 Training Method

Deep Spiking Neural Networks (SNNs) are typically trained using ANN-to-SNN conversion or direct training. While ANN-to-SNN conversion faces latency challenges, direct training is more time-step efficient and suitable for temporal tasks. We use rate coding for its support of complex SNNs. In this paper, we use the SuperSpike [73] surrogate gradient to calculate gradients, with detailed explanations provided in appendix A.7.

## 5 Experiment

### 5.1 Gesture Recognition based on sEMG

We compared our model's performance with existing sEMG-based gesture estimation models, primarily categorized into DNN and SNN architectures. A comparison summary is in Table 1.

In terms of Top-1 Accuracy, our JASNN model, which integrates the SNN framework with the SJA mechanism, outperforms other DNN models, including CNN, TCN [5], Transformer [60], GRU [11], Informer [80], and a hybrid TCN with an Attention mechanism. This superior performance is due to: 1) The SNN structure's alignment with the biological basis of sEMG generation, providing a natural

Table 1: Comparison with previous works on sEMG-based gesture estimation.

| Methods | Work | Model | Top-1 Acc.(%) | Std. Dev. (%) |
|---|---|---|---|---|
| DNN | Asif *et al.* 2020 [3] | CNN | 75.46 | 0.52 |
| | Tsinganos *et al.* 2020 [59] | TCN | 79.69 | 0.83 |
| | Rahimian *et al.* 2021 [47] | Transformer | 84.23 | 0.37 |
| | Chen *et al.* 2021 [9] | GRU | 82.19 | 0.28 |
| | Zhou *et al.* 2021 [80] | Informer | **88.32** | 0.36 |
| | Rahimian *et al.* 2022 [48] | TCN+Attention | 87.10 | 0.57 |
| | Zhang *et al.* 2023 [78] | Transformer | 86.24 | 0.31 |
| SNN | Bellec *et al.* 2018 [7] | LSNN | **86.24** | 0.22 |
| | Zhang *et al.* 2022 [74] | SIB+SNN | 77.84 | 0.62 |
| | Xu *et al.* 2023 [63] | SCNN | 84.30 | 0.23 |
| SNN-Ours | SOTA backbone [7] | LSNN+SJA(Ours) | 88.10 | 0.25 |
| | **This Work** | JASNN | **89.26** | 0.31 |

modeling of the processes. 2) The SJA mechanism's enhancement of sparse spike train features focuses on the key characteristics of sEMG signals. Compared to other SNN models like LSNN [7], SIB+SNN [78], and SCNN, our model achieves higher accuracy. Models like SIB+SNN and SCNN perform lower, likely due to the absence of a feature enhancement design like SJA, which is crucial for capturing the temporal dynamics of sEMG signals. Incorporating SJA into Xu *et al.*'s LSNN network [63] significantly improved performance, demonstrating SJA's scalability in recurrent SNNs.

## 5.2 Ablation Study

To validate the effectiveness of each module we have proposed, we present the results of an ablation study. Here, we discuss the impact of the backbone's attention mechanisms and loss functions on the experimental results and the influence of different pseudo-label generation methods within SSFDA. All experimental results in this section are based on the mean values across all fifteen subjects in the dataset. The same learning rate, batch size, and optimizer were used during training, ensuring each network converges (with training set accuracy showing less than $0.2\%$ improvement over five consecutive epochs). This thorough examination allows us to isolate the individual contributions of the different components and clarify their specific roles in the performance of our proposed system.

### 5.2.1 Attention Mechanisms

In our ablation study on attention mechanisms, we compared Raw Attention [60], MA-SNN [67], and our proposed Spiking Jaccard Attention (SJA) on SCNN, keeping all other parameters consistent. As shown in Table 2, using Raw Attention directly on spikes resulted in an accuracy of only 11.31% due to the high sparsity of spike sequences. This sparsity often leads to information loss when multiplying matrices with sparse values. MA-SNN converts spike sequences into continuous values and uses a fully connected layer for attention, which increased SCNN's accuracy from 84.12% to 85.67%. However, this approach reduces the usability of SNNs on spiking chips. In contrast, our SJA computes attention weights directly on the spike sequence, preserving compatibility with spiking hardware and further boosting accuracy to 87.44%. This highlights SJA's superior ability to handle spike sequence sparsity while maintaining hardware compatibility.

### 5.2.2 Loss Functions

The study compared Negative Log Likelihood (NLL) loss and our improved Smooth NLL with Kullback–Leibler divergence loss (SNLL+KLL) for classification tasks. Using SNLL+KLL in JASNN increased accuracy from 87.44% to 89.72% (see Table 2). This enhancement is due to: **Kullback–Leibler (KL) divergence:** KL divergence quantifies the difference between two probability distributions, encouraging predicted probabilities to closely match actual class distributions. This reduces model biases towards certain categories. **Smooth NLL (SNLL):** SNLL ensures consistent predictions between a feature and its k-nearest neighbors in the embedding space, enhancing model sensitivity to detailed class clusters and underlying patterns. In summary, adding KL divergence and SNLL improves model strength and fairness, enhancing flexibility across various datasets and tasks.

### 5.2.3 Pseudo-Label Generation Methods

We evaluated three pseudo-label (PL) generation methods: Duan *et al.* [17] (PL), Huang *et al.* [28] (NPL), and our proposed Probabilistic Label Generation (PLG) method. Both PL and NPL improved

Table 2: Ablation study results

| | | Attentions | | | Loss Functions | | Label Selection | | | Results | |
|---|---|---|---|---|---|---|---|---|---|---|---|
| | SCNN | RA | MA-SNN | SJA (Ours) | NLL | SNLL+KLL (Ours) | PL | NPL | PLG (Ours) | ACC | Improved ACC |
| Backbone | ✓ | | | | ✓ | | | | | 84.12% | - |
| | ✓ | ✓ | | | ✓ | | | | | 11.31% | - |
| | ✓ | | ✓ | | ✓ | | | | | 85.67% | - |
| | ✓ | | | ✓ | ✓ | | | | | 87.44% | - |
| | ✓ | | | ✓ | | ✓ | | | | **89.72%** | - |
| Source-Free | ✓ | | | ✓ | | ✓ | ✓ | | | - | 1.87% |
| | ✓ | | | ✓ | | ✓ | | ✓ | | - | 2.33% |
| | ✓ | | ✓ | | | ✓ | | | ✓ | - | 3.81% |
| | ✓ | | | ✓ | | ✓ | | | ✓ | - | **4.10%** |

accuracy in an unsupervised setting on JASNN by 1.87% and 2.33%, respectively. Our PLG method achieved a significant boost, enhancing accuracy by 4.1%. Additionally, PLG increased accuracy on MA-SNN by 3.81%, demonstrating its scalability across different architectures. The effectiveness of PLG comes from selecting the mode of neighboring labels as the pseudo-label and introducing a probabilistic mechanism to explore other labels, preventing overly compact feature distribution and enhancing generalization. This makes PLG a powerful tool for unsupervised adaptation in sEMG-based gesture recognition.

## 5.3 Variant Distributions of Hand Gestures with Three Different Forearm Postures

In this study, we evaluated datasets from three forearm postures (P1, P2, P3) to train models without SSFDA, revealing challenges with sEMG data. The model trained on P1 achieved high accuracy on P1's test set but dropped to 30% accuracy on P2 and P3 due to variations in motor neuron firing patterns. Figure 9 illustrates this performance disparity, highlighting the out-of-distribution (OOD) issue. These findings underscore the need to address OOD phenomena in sEMG data to enhance the reliability and user experience of sEMG-based systems.

## 5.4 Result of Spiking Source-Free Domain Adaptation

In our investigation, we used the same dataset to experiment with three different methodologies, namely Pseudo-Label [34, 17] method, Neighborhood-guided Pseudo-Labels [28] method, and our proposed Probabilistic Label Generation method. In this experiment, the Pseudo-Label method determines the pseudo-label by taking the mode of the $k$ samples. Conversely, the neighborhood-guided Pseudo-labeles method involves choosing the nearest $k$ samples from the memory list and then randomly selecting one from these $k$ samples as the pseudo-label. Details of our proposed method have been elaborated on in the previous sections of the paper.

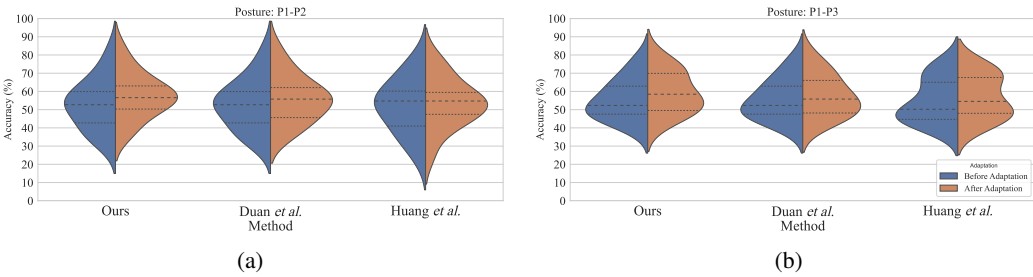

Figure 4: Comparison of performance before and after applying SSFDA for various methodologies: Figures 4a and 4b are Violin Plots demonstrating this disparity.

Figure 4a and 4b represent the performance variations when deploying the model trained on Posture 1 to Posture 2 and 3, respectively, both with and without the use of our SSFDA. This is portrayed using a violin plot. It can be observed that the use of SSFDA indeed shifts the distribution of accuracy across different subjects upward as a whole. Particularly, our method exhibits superior performance after applying SSFDA compared to the other two methods. Furthermore, the standard deviation of performance across various subjects is minimal for our method, demonstrating the robustness of our methodology when employing SSFDA. We detailed the differences by individuals in Figure 10a 10b in the appendix.

## 5.5 Efficiency Analysis of Spiking Jaccard Attention

Inference latency influences user experience, with delays leading to missed or inappropriate actions. Attention mechanisms are pivotal for efficient interactions. We compared SJA's computational superiority over Raw [60] and Efficient [54] Attention, conducting 100 inference tests using pseudo-data on twelve channels, a common practice in sEMG data. Results averaged and shown in Figure 5, highlight SJA's clear advantages. Regardless of the computing platform or data type, SJA demonstrated superior efficiency in inference speed and RAM consumption, making it ideal for real-time and mobile devices. Additionally, SJA showed better scalability, with only a gentle increase in inference time and RAM usage as sEMG data length increased, compared to Raw Attention's exponential growth.

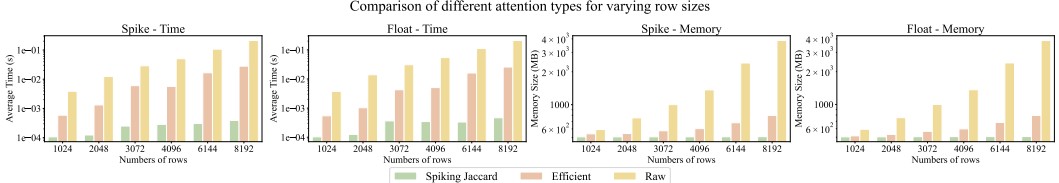

Figure 5: Inference speed and RAM usage comparison between spike and float data for Raw Attention [60], Efficient Attention [54], and our Spiking Jaccard Attention: The first column shows inference time for float data, and the second for spike data. The third and fourth columns show RAM usage for these data types. The $x$-axis represents different data row counts, and the $y$-axis is logarithmic to highlight performance differences. Each experiment was conducted 100 times, with averaged results.

## 5.6 Real World Deployment

SpGesture has been deployed in a real-world application using an in-house developed sEMG acquisition system, as illustrated in appendix A.10.

## 6 Limitation and Future Work

**Domain Adaptation on Various Network Structures:** We verified the ability of SJA and SSFDA to enhance the accuracy of sEMG-based gesture recognition, along with their adaptability to distribution shift based on the Spiking Convolutional Neural Network architecture. Moving forward, we intend to assess their robustness across a wider variety of SNNs and different tasks.

**Performance Analysis on Neuromorphic Chips:** Our current evaluations of inference speed and memory utilization are conducted on CPU and GPU platforms, where our system demonstrates clear advantages over existing algorithms. We believe that these advantages will be further amplified on neuromorphic chips. We are currently developing neuromorphic chips and will conduct practical tests on these chips to measure the system's energy consumption and inference efficiency.

## 7 Conclusion

We presented SpGesture, an innovative framework for sEMG-based gesture recognition built on SNN, and innovatively introduced Spiking Source-Free Domain Adaptation with Spiking Jaccard Attention, which directly enhances spike features. These novel contributions improve the system's robustness and accuracy in real-world scenarios. Our experimental results include the highest accuracy among baselines and system latency below 100ms on a CPU, demonstrating its real-world applicability. Our proposed SJA processes spike sequences at 36.37 times the speed of conventional attention and can be extended to other SNNs, such as LSNN. SpGesture not only offers a practical solution to current challenges in gesture recognition but also opens new possibilities for Human-computer Interaction.

## Acknowledgments and Disclosure of Funding

This work is partly supported by the National Key Research and Development Program of China (No. 2023YFF0725001), the National Natural Science Foundation of China (No. 92370204, 62306255), the Natural Science Foundation of Guangdong Province (No. 2024A1515011839), the Guangzhou-HKUST(GZ) Joint Funding Program (No.2023A03J0008), and the Education Bureau of Guangzhou Municipality. Yijie Xu acknowledges the support from the modern matter laboratory, HKUST(GZ).

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

# A    Appendix / supplemental material

## A.1    Details of Data Collection

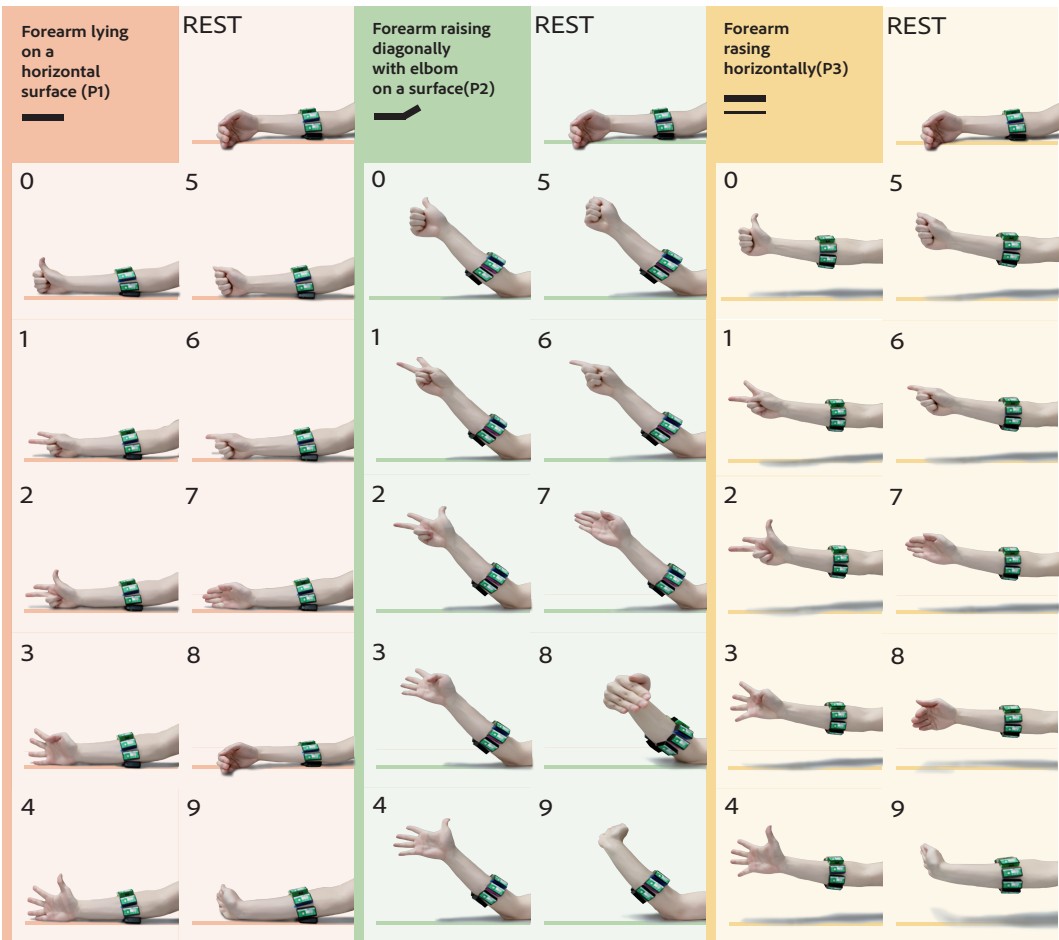

Figure 6: Overview of our dataset: the compilation contains sEMG data for ten distinct actions, each across three postures. Varied background colors represent distinct forearm postures, while the digits ranging from 0 to 9 correspond to specific gesture actions. The 'Rest' label at the top denotes a static hand gesture when no action is being performed.

In human-computer interaction studies focusing on surface electromyography (sEMG), the acquisition of diverse and representative data sets is crucial. Current research predominantly collects sEMG data from gestures made with a single forearm posture [4, 31, 46, 43, 32, 33, 29]. However, it is evident that variations in forearm posture can significantly influence the distribution of the sEMG data, potentially causing discrepancies between laboratory results and real-world applications. To address this, our data collection methodology incorporates gestures performed in different forearm postures, aiming to reflect the conditions and variability encountered in practical scenarios more accurately.

The experiments were carried out using the DataLITE wireless LE230 and DataLITE PIONEER, commercial sEMG acquisition systems from Biometrics Ltd.* The device's sampling rate is 2000Hz, allowing for high-resolution data capture of the electrical activities in the muscles during the performance of gestures. Eight LE230 sEMG sensors were uniformly and equidistantly affixed to the surface of the participant's right forearm.

---

*https://www.biometricsltd.com/

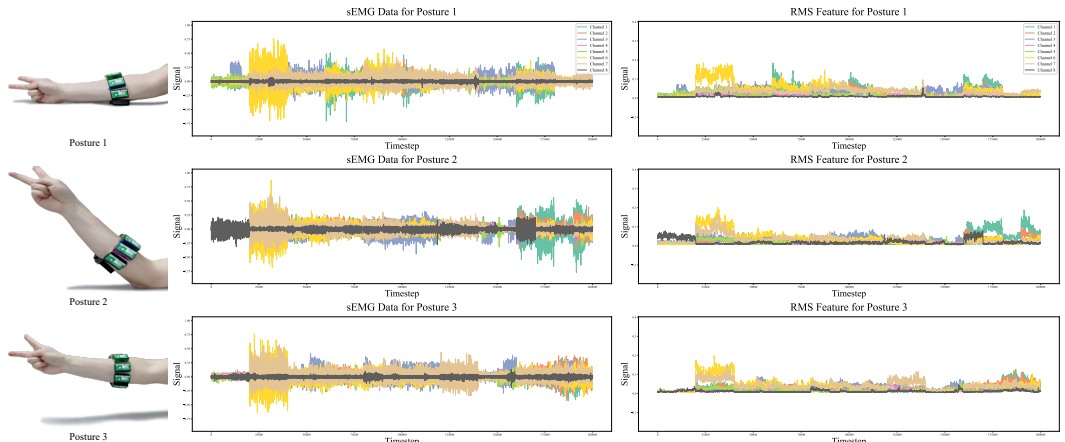

Figure 7: Summary of our collected data: The three postures on the left illustrate distinct preparatory arm actions. The central data graph represents the sEMG data captured from a subject under the three postures, with unique colors assigned to different channels. The data graph on the right showcases the acquired data after Root Mean Square (RMS) processing.

A total of fifteen subjects participated in the dataset, comprising ten males and five females. A significant proportion of our participant pool (twelve individuals) was right-handed, while three individuals were left-handed. None of the participants had any neurological or muscle disorders, ensuring the generalizability of our results to a healthy population. This study has been approved by the relevant university ethics committee, and it's worth noting that none of the participants had prior experience in using sEMG collection devices.

Before the experiment, each participant was thoroughly briefed about the procedures. They willingly participated in the study, aware that their data would be open-source. During the experiment, the subjects were instructed to replicate the gestures and forearm postures displayed on a screen using their right or left hand. This methodology allowed us to systematically record sEMG signals from the participants across a range of forearm positions and gestures.

Most existing datasets for surface electromyography (sEMG) gesture recognition predominantly use a single forearm posture, which does not align with real-world scenarios, where forearm posture varies dynamically and significantly influences the muscle state. As shown in Figure 7 and Figure 9, changes in forearm posture lead to alterations in the distribution of sEMG signals, resulting in a substantial decline in the prediction accuracy of gesture estimation models. Acknowledging this limitation in prevailing datasets, our study aims to bridge this gap. In our study, we incorporated three distinct forearm postures for our data collection, as depicted in Figure 6. In the study, three distinct forearm postures were identified: P1, where the forearm was placed horizontally on a flat surface; P2, where the forearm was elevated diagonally with the elbow anchored on a surface; and P3, where the forearm was maintained in a horizontal orientation. These configurations are crucial for understanding ubiquitous computing interactions related to forearm ergonomics. Our gesture set included ten daily-life actions: Thumb up, Extension of the index and middle fingers, Flexion of the others; Flexion of the ring and little finger, Extension of the others; Thumb opposing base of little finger; Abduction of all fingers; Fingers flexed together in a fist; Pointing index; Adduction of extended fingers; Wrist flexion and Wrist extension. During the experiment, participants were instructed to maintain one of the specified gestures for five seconds, followed by a five-second relaxation period. This process was repeated six times for each gesture under each forearm posture. This methodology, combining varying forearm postures and gestures, aims to provide robust and diverse sEMG data. In the future, we will collect more distribution shift scenarios, like the Electrode movement.

## A.2 Details of Data Preprocessing

In the data preprocessing phase, we utilized the Root Mean Square (RMS) as the initial feature extraction method to improve the stability of gesture recognition. RMS serves as an advantageous choice for feature extraction due to its ability to efficiently summarize the magnitude of the signal

variation, providing a stable indication of the signal power. To capture the transient characteristics of the sEMG signals in our study, we used a time window length of 100ms with a step size of 0.5ms for the RMS calculation. This approach allowed us to extract representative RMS features from the signal while maintaining a high-resolution view of the signal variations. RMS can be mathematically represented as follows:

$$\mathrm{RMS}\left(\mathbf{X}\right) = \sqrt{\frac{1}{N}\sum_{i=1}^{N} x_i^2}, \tag{11}$$

where $x_i$ represents each value in the signal $\in \mathbf{X}$, and $N$ is the total number of values or samples in the signal.

## A.3   Spiking Neural Networks

Spiking Neural Networks (SNNs) represent the third generation of neural networks [41, 21, 20] and are advantageous in several key aspects when compared to traditional Deep Learning (DL) models. For instance, they inherently handle temporal information [19, 66, 81], efficiently process event-based data [82], and offer lower energy consumption for certain tasks [24] due to their sparse [45, 10] and asynchronous nature [61]. Emulating the precise mechanism of neuronal spike transmission in the human brain, SNNs stand at the frontier of biologically inspired artificial intelligence [72], offering potential advancements in neuromorphic computing [18] and beyond. SNN processes information and generates 'spikes' or 'impulses' only when a certain or dynamic threshold of neuron activation is reached, leading to a more efficient representation and transmission of information. Fundamentally, sEMG signals are generated by neural impulses, making them naturally compatible with the processing mechanism of SNNs. The spikes in an SNN represent discrete events in time, which parallels the nature of sEMG signals containing valuable information in spatial (across different muscles) and temporal (over time) dimensions. This innate alignment between SNNs and sEMG data enables the efficient decoding of intricate patterns for gesture recognition. Furthermore, SNNs have lower power consumption compared to traditional DL models [18]. This feature aligns with the typical use case of sEMG in wearable technology, where power efficiency is a crucial consideration [38]. Traditional DL models require substantial computational resources for training and inference, but SNNs operate in an event-driven manner, only processing data when a spike occurs [55]. This unique attribute makes SNNs a feasible solution for real-time, low-power wearable applications [25], providing an effective solution for practical constraints in human-machine interaction systems.

While SNNs show promise for sEMG-based gesture recognition, they face challenges, such as lower accuracy [58] due to sparser feature representation and difficulties in convergence because of the lack of a natural gradient [15]. To mitigate the issue of sparse features, we propose integrating an attention mechanism into our SNN model, allowing it to focus on more relevant features within the sEMG data. This combined approach aims to enhance the accuracy and training effectiveness of SNNs for gesture recognition tasks.

## A.4   Conv-based Leaky Integrate-and-Fire Neurons

To transform the input into a spike train and learn the spike representation of sEMG samples, we use a smaller quantity of Conv-based Leaky Integrate-and-Fire (LIF) neurons. As shown in Figure 8a, Conv-based LIF neurons can process spatial and temporal features concurrently. Since sEMG signals consist of pulses from multiple motor neurons at the skin's surface combined with noise, using LIF neurons is biologically plausible for decoding these signals into spikes. Conv-LIF neurons also constrain the decoding range within local time, mitigating the impact of noise on global decoding. This method aligns with the biological characteristics of sEMG signals and effectively manages noise interference.

Initially, convolution is used to extract spatial features $S^{t,n}$ from the present spatial input $X^t$. These extracted features are then combined with the temporal features $T^{t-1,n}$ of the previous moment to form the current membrane potential $M^t$. Following this, the Fire and Leak module, based on the current membrane potential, generates the temporal features $T^{t,n}$ for the subsequent moment and the spatial features $S^{t,n+1}$ for the next layer of the network. A significant event, known as a spike, occurs if the membrane potential $M^t$ exceeds a threshold value, denoted as $V_{th}$. In such an event, $M^t$ is reset to a value $V_{\mathrm{reset}}$, and $T^{t,n}$, the post-synaptic potential, is calculated as the product of $S^{t,n+1}$ and

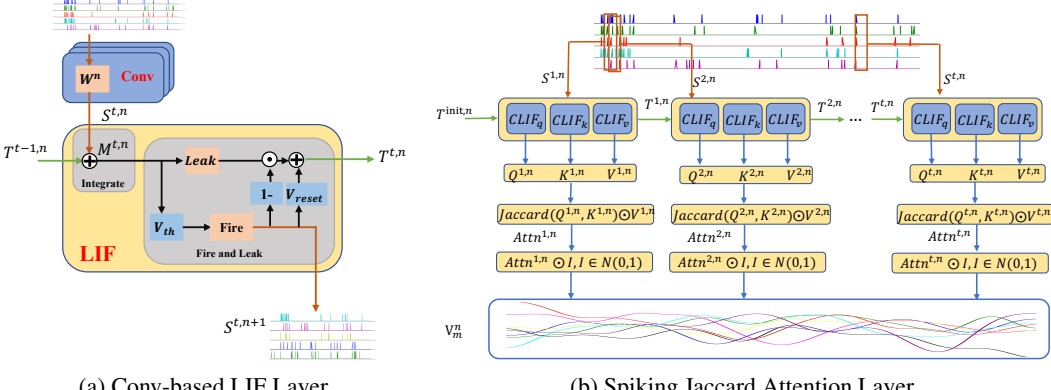

| (a) Conv-based LIF Layer | (b) Spiking Jaccard Attention Layer |
|---|---|

Figure 8: Illustrations of different layers in the network.

$V_{\text{reset}}$. On the contrary, if the membrane potential $M^t$ is lesser than the threshold $V_{th}$, a spike does not occur. In this case, $T^{t,n}$ is determined by $M^t$ and is computed as the product of the time constant $e^{-dt/\tau}$ and $M^t$.

Given that sEMG is composed of the summation of pulses produced by multiple motor neurons at the skin's surface, combined with a series of noise, it is biologically plausible to use Leaky Integrate-and-Fire (LIF) neurons, similar to motor neurons, to decode sEMG into spikes. Concurrently, Conv-LIF can constrain the decoding range within local time, avoiding the impact of noise on global decoding. This approach aligns with the biological characteristics of the signals and provides a robust method for managing noise interference.

## A.5   Complexity Analysis

Next, we will analyze and compare the complexity of traditional attention and SJA. The traditional attention mechanism consists of several operations that contribute to its computational complexity. Firstly, the dot product $QK^T$ is computed, which has a complexity of $O(n^2 \cdot d)$ where $n$ is the sequence length and $d$ is the dimensionality of queries, keys, and values. The next operation is scaling the dot product by $1/\sqrt{d_k}$, which requires $O(n^2)$ operations. Following this, the softmax function is applied. Computationally, this operation also requires $O(n^2)$ operations because, for each element, we must sum over all other elements to normalize them. Finally, we multiply the resulting matrix with the value matrix V. This operation has a complexity of $O(n^2 \cdot d)$ as well. Therefore, the overall time complexity of the attention mechanism is $O(n^2 \cdot d)$ due to the complexity of the matrix multiplication steps. Moreover, the space complexity is $O(n^2)$ to store the attention weights for each token pair in the sequence, which can be particularly costly for long sequences. This quadratic dependency on the sequence length $n$ is one of the main computational challenges of attention mechanisms.

On the other hand, the computational complexity of SJA depends on the number of non-zero elements in the vectors, owing to the sparse nature of SNN outputs. This sparsity allows us to focus our computation only on the non-zero elements, thus reducing the complexity. Specifically, for both intersection and union calculations, we perform minimum and maximum operations, respectively, between each pair of corresponding elements in vectors x and y. Given the sparsity, the complexity for both these operations is $O(b)$, where $b$ stands for the number of non-zero elements. Additionally, the summation operation in the numerator and the denominator of the Jaccard similarity formula also has a complexity of $O(b)$ because we are adding up the number of non-zero elements. Hence, the overall time complexity of the Spiking Jaccard Attention mechanism is $O(b)$. This is significantly more efficient than the traditional attention mechanism, especially in the context of SNNs, where the output is predominantly sparse.

This reduced complexity, while maintaining effective attention functionality, highlights the advantage of our proposed SJA mechanism for SNNs, enhancing their computational efficiency without compromising on performance. Additionally, the energy consumption of the traditional attention dominated by multiplication could be several times higher than that of the SJA dominated by addition.

For example, the energy cost of a multiplication ($3.7 \, pJ$) is $4.1\times$ to an addition ($0.9 \, pJ$), in 45nm CMOS technology [27].

## A.6 Details of LIF-based Classifier

The Leaky Integrate-and-Fire (LIF) model [83] provides a biologically plausible and computationally efficient approximation of neuronal spike generation. We employ a modified Leaky Integrate-and-Fire (LIF) layer as the classifier and add a memory module to record the membrane potentials, which receive spikes from the preceding network and translate them into membrane potentials. The number of LIF neurons corresponds to the task's categories, and the neuron with the highest membrane potential determines classification. This approach provides an efficient method for translating spiking activity into classification results.

Mathematically, the LIF model is represented by the following differential equation:

$$\tau \frac{dV(t)}{dt} = -V(t) + RI(t). \tag{12}$$

Here, $V(t)$ characteristically denotes the membrane potential of a neuron, $R$ represents the inherent membrane resistance, $I(t)$ is the incoming current or the external input signal, and $\tau$ is the vital time constant of the neuron, which is fundamentally the product of the membrane resistance and its capacitance.

When the incoming signals ($I(t)$) cause the membrane potential ($V(t)$) to exceed a predefined threshold ($\theta$), the neuron 'fires' a spike, and then its membrane potential is reset to a resting potential. The 'leaky' aspect comes from the $-V(t)$ term in the equation, which models the neuron's natural decay towards the resting potential in the absence of input, effectively avoiding an unbounded increase of membrane potential. Striking a balance between computational simplicity and biological characteristics, the LIF model is computationally less demanding, making it suitable for wearable, real-time applications such as sEMG-based gesture recognition. Moreover, its inherent ability to handle time-series data is critical in capturing the temporal dynamics of sEMG signals.

The following equations provide a straightforward iterative formulation for the LIF-SNN layer, facilitating easier inference and training processes:

$$\begin{cases} M^t = T^{t-1} + X^t, \\ S^t = Hea\left(M^t - u_{\text{th}}\right), \\ T^t = V_{\text{reset}}S^t + \left(e^{-\frac{dt}{\tau}}M^t\right) \odot \left(1 - S^t\right). \end{cases} \tag{13}$$

In these equations, $M^t$ represents the membrane potential at the $t$-th time step. $X^t$ represents the spatial feature and $T^{t-1}$ represents the temporal feature. $\odot$ stands for element-wise multiplication. When $M^t$ reaches or exceeds the threshold $u_{\text{th}}$, it is reset to $V_{\text{reset}}$, and a spike ($S^t = 1$) is emitted. Otherwise, $M^t$ evolves according to the given dynamics, and no spike is emitted ($S[n] = 0$). $e^{-dt/\tau} < 1$ stands for the decay factor.

## A.7 Details of the Training Method

Deep Spiking Neural Networks (SNNs) are typically trained using two methods: ANN-to-SNN conversion and direct training. ANN-to-SNN conversion approximates ANN activation values with SNN firing rates. However, it involves a trade-off between accuracy and latency, requiring sufficient time steps for accurate rate-coding. Despite its application in large-scale structures like VGG and ResNet, it faces challenges in latency, restricting its practicality. Direct training of SNNs, on the other hand, applies continuous relaxation of non-smooth spiking for backpropagation. It outperforms ANN-to-SNN conversion in time step efficiency and is suitable for temporal tasks. Though it can employ various coding schemes, we choose rate coding in this paper for its ability to support complex SNNs. The SuperSpike [73] surrogate gradient can be described as:

$$\sigma'\left(U_i\right) = \left(1 + |U_i - v|\right)^{-2}. \tag{14}$$

The formula is designed in such a way that the contribution to the gradient becomes significant when the neuron's membrane potential ($U_i$) is close to the firing threshold ($v$). In the case when $U_i$ is much

higher than $v$, implying the neuron is certain to fire, the contribution of the neuron to the gradient is lessened. This is because the neuron's firing state is unlikely to be affected by small changes. Hence, it is not crucial for the current learning step. Conversely, the gradient contribution remains low if $U_i$ is much lower than $v$, indicating the neuron is less likely to fire. This is because it would take a substantial adjustment to this neuron's activity to make it fire, suggesting it currently has a minimal impact on the network's overall output.

Hence, the SuperSpike algorithm optimizes the learning process by focusing on the neurons that are on the verge of changing their firing status, i.e., when $U_i$ is close to $v$. This ensures the resources are directed towards neurons that can be effectively adjusted by the current learning step, improving the overall learning efficiency.

## A.8 Implementation Details

Our model was developed using PyTorch and Norse [44]. Norse is a library that extends PyTorch to support the development of Spiking Neural Networks (SNNs). We trained our model on a server with two AMD EPYC 7543 32-core Processors, one NVIDIA RTX 3090 GPU, and 1TB RAM. To ensure model convergence, we iterated over 200 epochs. Model convergence was determined by criteria wherein if the accuracy improvement was less than $0.2\%$ within ten epochs, the model was deemed to have reached a convergent state.

The model was trained using the Adam optimizer, a popular choice for deep learning tasks due to its ability to handle sparse gradients on large-scale datasets and adapt the learning rate based on the computation of adaptive learning rates for different parameters. The learning rate was initially set to $0.001$, and the batch size was set to 32. However, to guarantee a fair comparison across all tested models, we train these models using the same parameters as the original papers.

In terms of data preparation and division, we maintained the same training-test set split across all datasets. Specifically, $70\%$ of the data was used for training, and the remaining $30\%$ was used for testing. We ensured no intersection between the training and testing sets to prevent data leakage.

The sEMG data was segmented using a window length of 100ms. To increase the number of training instances and capture the transitional states between actions, we applied a $50\%$ overlap between adjacent samples.

Finally, to ensure the fairness of the comparison, all the models were trained based on the Root Mean Square (RMS) features. This was performed regardless of the feature extraction methods initially proposed in their papers. This approach allows a more accurate comparison of the performance of the different models.

## A.9 Source-free Domain Adaptation on Out-of-distribution Data

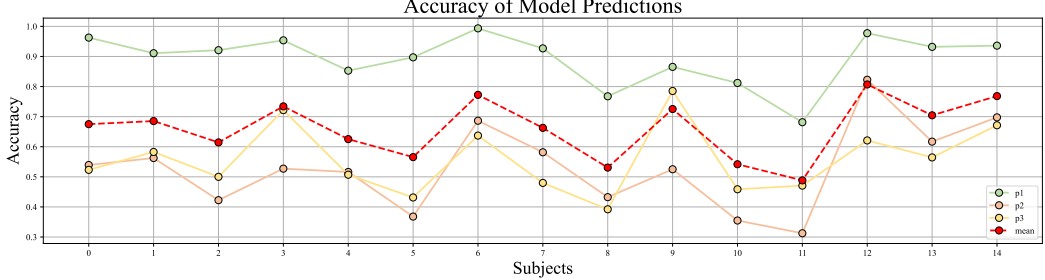

Figure 9: Demonstration of our dataset's inherent out-of-distribution (OOD) nature: We used data from Posture 1 for inference on data from Postures 1, 2, and 3 and subsequently calculated the accuracy. This highlights the OOD characteristics of data with different pre-existing postures.

Figures 10a and 10b illustrate the performance variations across different subjects when the model trained on Posture 1 is deployed on Posture 2 and 3, respectively, after using our SSFDA. It is noticeable that our method consistently yields improved performance instead of deterioration. In the majority of the subjects, the accuracy improvement of our SSFDA is better than the other two

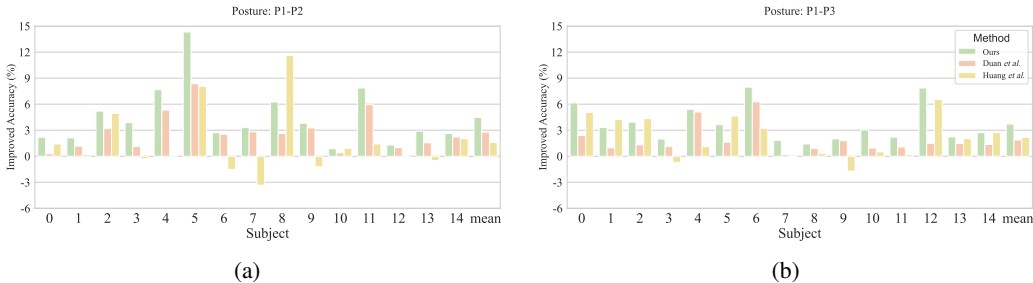

(a)                             (b)

Figure 10: Comparison of performance before and after applying SSFDA for various methodologies: Figures 10a and 10b present bar charts depicting the difference in performance before and after SSFDA for different subjects using three methods.

strategies. Additionally, our average accuracy improvement is significantly higher than that of PL and NL. This indicates that our proposed Probabilistic Label Generation method is more conducive to learning a distribution with generalization capabilities than PL and NL.

## A.10 Real World Deployment

SpGesture has been deployed in a real-world application using an in-house developed sEMG acquisition system, as illustrated in Figure 11. The system comprises analog front-end sensing circuits (AFE), a microcontroller (MCU), and a wireless module. The system is equipped with eight AFE channels that can be attached to the surface of the forearm. Dry electrodes and instrumentation amplifiers are utilized to sense and amplify the sEMG signals, respectively. A 32-bit MCU with ARM Cortex-M4 is employed to control the acquisition system, and the eight-channel amplified sEMG signals are sampled by a 12-bit analog-to-digital converter (ADC) inside the MCU at a sampling rate of 2000Hz. The converted digital signals are wirelessly transmitted to a computer host through a low-power Bluetooth (BLE) 5.2 module. The acquisition system is powered by a Li-ion battery.

We have tested our JASNN and SSFDA algorithms on this hardware device, and the results are encouraging. Even under the constraint of utilizing only the CPU, our inference latency was found to be less than 100ms. This swift response time falls within the real-time requirements for most application scenarios. Such low latency, coupled with the carefully designed acquisition system, demonstrates that our approach is not only theoretically sound but also practically viable for real-world deployment in various sEMG-based applications.

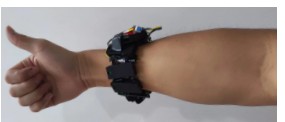 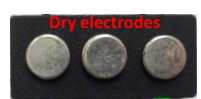 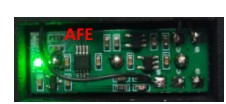 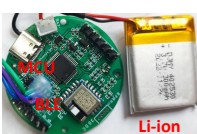

Figure 11: In-house developed sEMG acquisition system: from left to right are the acquisition system, dry electrodes, AFE, MCU with BLE 5.2 module, respectively.

## A.11 Instructions to participants of research with human subjects

In our study, participants will engage in an interactive hand movement activity, which is detailed as follows:

- **Viewing the Movements:** Participants will watch a series of short videos on a laptop screen, each showing a specific hand movement.
- **Repeating the Movements:** After viewing each video, participants will mimic the hand movement shown using their right hand. This includes various gestures and movements as outlined in hand movement taxonomies and robotics literature.
- **Equipment Setup:** Participants will wear a special glove (dataglove) and an accelerometer attached to their wrist. These devices will record the kinematic information of the hand movements.

- **Muscle Activity Recording:** We will attach 8 to 12 wireless electrodes to each participant's forearm to measure muscle activity. These electrodes will be placed at specific locations around the forearm and on key muscle areas, like the biceps and triceps, following an anatomically informed strategy for precise data collection.
- **Ensuring Comfort and Stability:** The electrodes will be secured using standard adhesive bands and a hypoallergenic, latex-free elastic band to ensure they stay in place throughout the activity.
- **Performing the Task:** Participants will be asked to repeat each demonstrated movement for 5 seconds, followed by a 3-second rest. This sequence will be repeated 6 times for each of the 49 different movements.

