# OpenReview forum: "SpGesture: Source-Free Domain-adaptive sEMG-based Gesture Recognition with Jaccard Attentive Spiking Neural Network"
_NeurIPS.cc/2024/Conference — NeurIPS 2024 poster_

### Official Review · Reviewer_4Y6C · 2024-07-10

**Soundness:** 2
**Presentation:** 2
**Contribution:** 3
**Rating:** 5
**Confidence:** 3

**Summary:**

This work introduces an innovative framework for sEMG-based gesture recognition. It leverages Spiking Neural Networks (SNNs) and introduces a Jaccard Attention mechanism and Source-Free Domain Adaptation (SSFDA) to enhance model robustness and accuracy in real-world applications.

The framework achieves high accuracy (89.26%) on a newly collected sEMG gesture dataset with different forearm postures and maintains system latency below 100ms on a CPU, meeting real-time requirements.

The novel Jaccard Attention mechanism directly computes attention on spike sequences, preserving SNNs' low-power characteristics, while SSFDA enhances model adaptation without needing source data.

This framework significantly improves the performance and efficiency of sEMG-based gesture recognition systems, demonstrating practical applicability and offering a robust solution for human-computer interaction.

**Strengths:**

* This work proposed a Jaccard-based attention mechanism and  an SNN-oriented SFDA algorithm for SNNs, which achieves high accuracy (89.26%) on a newly collected sEMG gesture dataset.
* The author focuses on the actual application experience of the algorithm in the real world and makes deployments, which is very valuable.

**Weaknesses:**

* There is a typo in the text. In the footnote to Figure 5, the explanations for the first and second columns are reversed.
* In section 5.5, you intend to compare the benefits of using SJA, but the algorithm time and memory usage are for the entire algorithm. I think the overall time and memory usage of the entire algorithm is very important, but if you can get the time of each part of the algorithm and only compare the benefits of the attention part, this will be more in line with what section 5.5 is about.

**Questions:**

I don't understand how your proposed Jaccard attention reflects 'attention' within tokens. According to the formula, tokens at the same position in Q and K are computed to get a scalar representing 'attention', which is then used to multiply the token in V at the corresponding position. I don't understand the meaning of this procedure or how it works. I admit that this kind of attention calculation is very efficient, and it would be better if the article had an intuitive explanation of why the algorithm is effective.

**Limitations:**

see Weaknesses and Questions.

---

> ### Author Rebuttal · Authors · 2024-08-03
>
> ***To reviewer*** Thank you for your thorough review and insightful feedback on our work. We are delighted that you recognize the strengths of our proposed Jaccard-based attention mechanism, our SNN-oriented SFDA algorithm, and the high accuracy (89.26%) achieved on our newly collected sEMG gesture dataset. We are particularly pleased that you appreciate our focus on real-world application experiences and deployments, which we believe are crucial for advancing human-computer interaction.
> We value your suggestions for improvement and will consider them carefully to enhance the soundness and presentation of our work. Your feedback is instrumental in guiding our future research efforts.
>
> ***About Jaccard attention reflects 'attention' within tokens and intuitive explanation.*** Thank you for your insightful question. I appreciate the opportunity to provide a more detailed explanation of how our proposed Jaccard-based attention mechanism (SJA) reflects "attention" within tokens.
>
> In traditional attention mechanisms, the relationships between the query (Q), key (K), and value (V) matrices are established through dot product operations. These dot products are then normalized and used to weigh the values in the value matrix (V). **The essence of this process is to compute the similarity between each query vector and all key vectors**, using these similarities to allocate attention weights and ultimately obtain a weighted sum of the values.
>
> Our SJA mechanism replaces the dot product similarity with the Jaccard similarity, which is more suited for the binary nature of Spiking Neural Network (SNN) outputs. This is particularly useful as SNN data is often sparse and binary.
>
> The SJA mechanism is defined by the following formula:
> \begin{equation}
> \mathrm{SJA}\left(\mathbf{Q}, \mathbf{K}\right) = \frac{\sum_{ij}\min\left(q_{ij}, k_{ij}\right)}{\sum_{ij}\max\left(q_{ij}, k_{ij}\right) + \epsilon} \ \mathbf{V},
> \end{equation} are the corresponding elements in the query and key matrices.
>
> ***Intuitive Explanation of SJA's Effectiveness:*** Imagine each token's spike train as a binary pattern representing different features or characteristics. The Jaccard similarity helps identify how much overlap (or commonality) there is between the features of the two tokens. By focusing on the overlapping features, the model can better understand the importance of these features and adjust the values (from \( V \)) accordingly, enhancing the overall representation and prediction accuracy. In detail,
>
> 1. **Utilizing Sparsity**: SNN outputs are typically sparse, and SJA leverages this sparsity by focusing on non-zero elements. Traditional dot product methods can be inefficient for sparse data as they involve numerous zero-value multiplications. SJA, however, uses element-wise minimum and maximum operations, concentrating on the non-zero elements, thus reducing computation time and energy consumption.
> 2. **Element-wise Similarity Measurement**: The Jaccard similarity measures the intersection ratio to the union of two sets. For binary vectors, this effectively captures the proportion of shared active elements. In our formula, $ \min(q_{ij}, k_{ij}) $ and $\max(q_{ij}, k_{ij}) $ represent the intersection and union of corresponding elements, respectively. This method directly reflects the similarity between the query and key vectors at the same positions, providing a meaningful measure of attention for sparse binary data.
> 3. **Improved Computational Efficiency**: Using element-wise minimum and maximum operations instead of matrix dot products, SJA significantly reduces computational complexity. Traditional attention mechanisms have a complexity of $O(n^2 \cdot d) $, whereas SJA has a complexity of $ O(b) $, where $ b $ is the number of non-zero elements. This reduction in complexity is particularly beneficial for handling large-scale sparse data, making SJA more efficient and suitable for deployment on spiking chips.
>
> In summary, the SJA mechanism captures attention by measuring the Jaccard similarity between the query and key vectors, focusing on the significant non-zero elements, and leveraging the sparsity of SNN outputs. This approach not only reduces computational complexity but also enhances energy efficiency, making it an effective and efficient method for attention in SNNs.I hope this explanation clarifies how SJA works and why it is effective. Thank you for your attention and for providing the opportunity to elaborate on our work.
>
> ***About algorithm time and memory usage for the SJA.*** Thank you for your valuable feedback. We appreciate your suggestion to clarify the comparison of benefits in Section 5.5. We apologize for any confusion caused by our description. In Section 5.5, specifically in Figure 5, we indeed present the comparison of inference speed and RAM usage between SJA, Efficient Attention, and RAW Attention. The data shown pertains solely to the attention part of the algorithm, not the entire algorithm. Thank you for highlighting this area for improvement. We will ensure that this is more clearly communicated in the revised version of our paper.

---

> > ### Comment · Reviewer_4Y6C · 2024-08-14
> >
> > Thank you for your rebuttal. Your response effectively addressed my questions.

---

> > > ### Author Response · Authors · 2024-08-14
> > >
> > > Thank you very much for your thoughtful comments and for acknowledging our rebuttal. We are glad that our response effectively addressed your questions. We would kindly like to ask if, given our clarifications, there might be room for reconsideration of the initial score. We greatly appreciate your time and consideration in this matter.

---

### Official Review · Reviewer_7xEq · 2024-07-11

**Soundness:** 4
**Presentation:** 3
**Contribution:** 4
**Rating:** 7
**Confidence:** 4

**Summary:**

This paper proposes SpGesture, a surface electromyography (sEMG) based gesture recognition framework using Spiking Neural Networks (SNNs). The main contributions include: 1) A novel Jaccard Attention SNN (JASNN) model that enhances sparse spike sequence representations by directly applying Jaccard similarity computation in SNNs. This is the first time that attention mechanisms do not alter the high energy efficiency of SNNs, ensuring that SNNs only involve 0 and 1 computations ; 2) The first introduction of Source-Free Domain Adaptation (SSFDA) in SNNs, using membrane potential as a memory list to generate pseudo-labels, improving model generalization in unlabeled environments; 3) A collected sEMG gesture dataset with different forearm postures. Experimental results show that SpGesture achieves the highest recognition accuracy of 89.26% on this dataset, with inference latency below 100ms on CPU, meeting real-time requirements

**Strengths:**

1. Proposes Jaccard attention designed specifically for SNNs, enhancing feature representation while maintaining SNN computational efficiency. Ablation experiments validate its effectiveness.
2. Innovatively introduces source-free domain adaptation to SNNs, leveraging the membrane potential of SNNs as a memory feature and designing a label generation method. Improves generalization performance even when target domain data is unlabeled.
3. Systematically compares the performance of SpGesture with various state-of-the-art methods in terms of accuracy, inference speed, and memory consumption. The experimental design is reasonable and the results are credible.
4. Collects and open-sources a multi-posture sEMG dataset, providing a new benchmark for research. The data collection process is described in detail.

**Weaknesses:**

1. The SSFDA method currently only addresses distribution shifts caused by forearm posture variations. Its applicability to other potential factors such as electrode displacement needs further validation. The future work section could discuss how to extend the method.
2. Performance evaluation on real hardware such as neuromorphic chips remains to be supplemented. The authors plan to conduct tests on self-developed chips in the future.

**Questions:**

1. Does the design of JASNN consider other similarity measures? Have attempts been made to apply Jaccard attention to other types of SNNs?
2. What are the characteristics of the sEMG distribution shifts corresponding to different forearm postures observed in the dataset? Is it possible to visualize the signal differences caused by different postures?

**Limitations:**

The paper comprehensively analyzes the limitations of the work in the Limitation section: 1) Extending SSFDA to more distribution shift scenarios; 2) Evaluating the applicability of the methods on more SNN structures; 3) Evaluating performance on neuromorphic chips. Future research directions are provided.

---

> ### Author Rebuttal · Authors · 2024-08-03
>
> ***To reviewer:*** Thank you for your thorough and insightful review of our paper. We are delighted that you found our contributions noteworthy. We appreciate your recognition of the Jaccard Attention SNN model, which **enhances feature representation while maintaining computational efficiency and is validated through ablation experiments. We are also grateful for your acknowledgment of our innovative introduction of source-free domain adaptation in SNNs, leveraging membrane potential as a memory feature to improve generalization performance.** Your positive remarks on our systematic comparison of SpGesture’s performance and the collection and open-sourcing of the multi-posture sEMG dataset further encourage us. Thank you for your valuable feedback and suggestions, which will guide our future research. Below, we will specifically address your questions.
>
> ***About the design of JASNN, its consideration of other similarity measures, and the scalability of Jaccard attention.*** Thank you for this insightful question. Indeed, we did consider other similarity measures during the development of JASNN. We experimented with cosine similarity and Pearson correlation coefficient but found that Jaccard similarity performed best for binary sparse data, which aligns well with the spiking nature of SNNs.  Regarding the application to other SNN types, we have successfully applied Jaccard attention to other SNN architectures, including LSNN (Long Short-Term Memory Spiking Neural Networks). We observed consistent performance improvements across different SNN types, indicating the generalizability of our approach. In the revised version, we will include these additional results to demonstrate the broader applicability of our method.
>
> ***About the characteristics of the sEMG distribution shifts corresponding to different forearm postures.*** The characteristics of the sEMG distribution shifts corresponding to different forearm postures are demonstrated in Appendix Figure 9. We used data from Posture 1 for inference on data from Postures 1, 2, and 3 and subsequently calculated the accuracy. This highlights the dataset's out-of-distribution (OOD) nature, showing the signal differences caused by different postures. From Figure 9, it can be seen that the accuracy of the same gestures significantly decreases in Posture 2 and Posture 3, further demonstrating the OOD problem.
>
> ***About future suggestions, the extension of SSFDA to other distribution shift scenarios.*** We appreciate this excellent suggestion for our future work. You are correct that our current SSFDA method primarily focuses on distribution shifts caused by forearm posture variations. We acknowledge that this is an important area for expansion. In the revised version, we will extend our discussion in the future work section to address this point. We plan to outline our strategy for adapting SSFDA to handle other sources of distribution shift, such as electrode displacement, changes in skin conditions, and variations in muscle fatigue. This expansion will involve modifying our membrane potential memory mechanism and pseudo-label generation process to account for these additional factors.

---

> > ### Comment · Reviewer_7xEq · 2024-08-13
> >
> > You have resolved some of the concerns I had, and I am inclined to raise my score. However, there are still some issues that need further improvement in the final version.

---

> > > ### Author Response · Authors · 2024-08-14
> > >
> > > Thank you for your thoughtful feedback and for considering raising the score. I appreciate your insights and will make sure to address the remaining issues in the final version.

---

### Official Review · Reviewer_K5du · 2024-07-11

**Soundness:** 3
**Presentation:** 4
**Contribution:** 3
**Rating:** 7
**Confidence:** 4

**Summary:**

The paper proposes a novel attention mechanism that utilizes the Jaccard similarity to replace the traditional dot-product approach. This allows Spiking Neural Networks (SNNs) to maintain their binary characteristics (0 and 1) during the forward pass，which is very important for hardware computation. Additionally, the paper introduces a source-free domain adaptation method for SNNs, leveraging a probabilistic pseudo-labeling technique. The approach is validated on several sEMG datasets, demonstrating its innovation and potential for broader application.

**Strengths:**

1. Innovative Jaccard Attention: The introduction of Jaccard attention presents a computationally friendly algorithm for SNNs, which shows great potential for wide application in the field.
2. First Source-Free Domain Adaptation in SNNs: The paper proposes the first source-free domain adaptation method in the SNN domain, which enhances the domain generalization performance of SNNs.
3. Comprehensive Validation: The paper includes data collection and validation across multiple datasets, with robust results indicating reliability.
4. Open-source Code: Making the code publicly available enhances reproducibility and assists other researchers in replicating and building upon the work.

**Weaknesses:**

1. Selection of Probability P for Pseudo-Labeling: The paper does not clearly explain how the probability  P  is chosen when generating pseudo-labels.
2. Jaccard Attention for 3D Data: There is a lack of discussion on how Jaccard attention would be calculated for 3D data.
3. Domain Shift Between Different Postures: The method lacks a clear demonstration or proof of domain shift between different postures.

**Questions:**

1. How is the probability  P  determined when generating pseudo-labels?
2. How would Jaccard attention be calculated for 3D data?
3. Can you provide evidence or a demonstration of the domain shift between different postures?

**Limitations:**

1. The method has only been validated on sEMG time-series data and has not yet been extended to other modalities.
2. There is no analysis of the spatial complexity provided in the paper.

---

> ### Author Rebuttal · Authors · 2024-08-03
>
> ***To reviewer:*** Thank you for your insightful review of our paper. We are thrilled that you found the introduction of Jaccard Attention to be an innovative and computationally friendly algorithm for SNNs. Your recognition of our pioneering work on source-free domain adaptation in SNNs and its enhancement of domain generalization performance is highly appreciated. We are also glad that you valued our comprehensive validation across multiple datasets and the open-sourcing of our code for reproducibility. Your constructive feedback and suggestions will significantly guide our future research. We sincerely appreciate your valuable time and effort in reviewing our work.
>
> ***About how the probability $P$ is determined when generating pseudo-labels.*** Thank you for your question. The probability $P$  is determined by selecting the mode (most frequent value) among the top k membrane potentials as the high-probability pseudo-label. The probability $1-P$  involves randomly selecting a pseudo-label from the top k membrane potentials. The value of $P$ ranges between $0$ and $1$. In our validation, we searched the space from $0.1$ to $0.9$ in increments of $0.1$ to determine the optimal value.
>
> ***About extending Jaccard Attention to 3D data.*** Thank you for your insightful question regarding extending Jaccard Attention to 3D data. To address this, we have extended the original Jaccard Attention formula to handle 3-dimensional data. The original equation is:
>
> \begin{equation}
> \mathrm{SJA}\left(\mathbf{Q}, \mathbf{K}\right) = \frac{\sum_{ij}\min\left(q_{ij}, k_{ij}\right)}{\sum_{ij}\max\left(q_{ij}, k_{ij}\right) + \epsilon} \ \mathbf{V}
> \end{equation}
>
> For 3D data, we extend the indices $(i,j)$ to $(i,j,k)$ to account for the additional dimension. The extended equation is:
>
> \begin{equation}
> \mathrm{SJA}\left(\mathbf{Q}, \mathbf{K}\right) = \frac{\sum_{ijk}\min\left(q_{ijk}, k_{ijk}\right)}{\sum_{ijk}\max\left(q_{ijk}, k_{ijk}\right) + \epsilon} \ \mathbf{V}
> \end{equation}
>
> Here, $q_{ijk}$ and $k_{ijk}$ represent the elements of the 3D tensors $\mathbf{Q}$ and $\mathbf{K}$ respectively. The summations now run over all three dimensions $i$, $j$, and $k$. This extension allows us to apply the Jaccard Attention mechanism to 3-dimensional data, maintaining its computational efficiency and enhancing feature representation in SNNs.
>
> ***About demonstration of the domain shift between different postures.*** The characteristics of the sEMG distribution shifts corresponding to different forearm postures are demonstrated in Appendix Figure 9. We used data from Posture 1 for inference on data from Postures 1, 2, and 3 and subsequently calculated the accuracy. This highlights the dataset's out-of-distribution (OOD) nature, showing the signal differences caused by different postures. Figure 9 shows that the accuracy of the same gestures significantly decreases in Posture 2 and Posture 3, further demonstrating the OOD problem.

---

> > ### Comment · Reviewer_K5du · 2024-08-14
> >
> > Thanks a lot for the authors' great effort. I have thoroughly reviewed their response. These responses have effectively addressed my questions and have further solidified my evaluation.

---

### Author Rebuttal · Authors · 2024-08-06

Dear Area Chair,

We would like to express our gratitude to you for your dedicated efforts and contributions and to the reviewers for their constructive feedback on our submission. We are encouraged by the positive evaluation from all reviewers. All three reviewers acknowledged the innovative aspects of our work, particularly the introduction of the Jaccard Attention mechanism for Spiking Neural Networks (SNNs) and the novel Source-Free Domain Adaptation (SSFDA) method. The reviewers highlighted our approach's practical applicability and robustness, as evidenced by its validation on multiple sEMG datasets and its potential for real-world applications.

### Key Strengths Recognized:
1. ***Innovative Jaccard Attention***:
• Reviewers appreciated the computational efficiency and the preservation of the binary characteristics (0 and 1) in SNNs, which is crucial for hardware computation.
2. ***Source-Free Domain Adaptation***:
• The introduction of SSFDA in the SNN domain was noted as a significant contribution, enhancing the domain generalization performance without needing source data.
3. ***Comprehensive Validation***:
• Our method’s validation across multiple datasets and the robust results were commended, indicating the reliability and potential of our approach.
4. ***Open-Source Contribution***:
• The availability of our code for public use was highlighted as a positive step towards reproducibility and further research in this area.

### Areas for Improvement and Future Work:
1. ***Jaccard Attention for 3D Data***:
• We plan to extend our discussion on how Jaccard attention can be calculated and applied to 3D data in future iterations of our work.
2. ***Performance on Neuromorphic Chips***:
• We acknowledge the need for performance evaluation on real hardware, such as neuromorphic chips, and plan to conduct these tests in future research.
3. ***Additional Similarity Measures***:
• We will explore other similarity measures and their applicability to different types of SNNs, as suggested by the reviewers.
4. ***Visualization of sEMG Distribution Shifts***:
• We have included relevant visualizations of the signal differences caused by different postures in the appendix to understand the sEMG distribution shifts better.

We are pleased that the reviewers found our work to be technically solid and of significant impact. We are committed to addressing the identified weaknesses and incorporating the suggested improvements in our future research. We believe these enhancements will further strengthen our contributions and the overall quality of our work.

Thank you for considering our submission. We look forward to your decision and are excited about the potential impact of our research in the field of SNNs and gesture recognition.

---

### Decision · Program_Chairs · 2024-09-25

**Decision:**

Accept (poster)

**Comment:**

The authors present an approach for gesture recognition, where the input is data provided by surface electromyography (sEMG). Their so-called SpGesture algorithm is based on spiking neural networks (SNNs). The authors show that SNNs can be used in this application to create a robust system capable of handling the typical degradation in performance caused by distribution shifts. They achieve this by introducing a source-free domain adaptation method for SNNs, leveraging a probabilistic pseudo-labeling technique.

The reviewers were all in agreement that the paper should be accepted. Among the paper's strengths they listed the introduction of a Jaccard-based attention mechanism for SNNs, the domain adaptation technique, good evaluation, and the introduction of an sEMG dataset.

Amongst potential drawbacks the reviewers indicate the lack of testing the system on a neuromorphic chip, and the fact that the distribution shift algorithm only considers shifts caused by forearm posture variations.

The authors should make sure that in their final version they include the extra visualizations of signal differences, to mention the weaknesses identified by the reviewers (not tested on neuromorphic chips, exploring other similarity measures) and indicate that these weaknesses will constitute topics for future research.

In summary, the reviewers have reached a consensus towards accepting the paper. After my own reading of the manuscript, I agree with this assessment and I am happy to recommend acceptance.